# Combination of Thermal and Mechanical Strategies to Compensate for Distortion Effects during Profile Grinding

Christian Schieber [1,*], Matthias Hettig [2,3], Michael Friedrich Zaeh [1] and Carsten Heinzel [2,3]

1   TUM School of Engineering and Design, Technical University of Munich, 85748 Garching, Germany
2   MAPEX Center for Materials and Processes, University of Bremen, 28359 Bremen, Germany
3   Leibniz Institute for Materials Engineering IWT, 28359 Bremen, Germany
*   Correspondence: christian.schieber@iwb.tum.de

**Abstract:** This paper describes the investigations and the results of an analysis of distortion compensation processes for profile grinding. Steel workpieces often change their residual stress state due to machining in a seemingly uncontrolled matter. Furthermore, in research as well as in the industry, the accurate representation of shape deviations during the cutting of slim profiled workpieces and their deformation handling is a major challenge. In this paper, a valid predictive model, developed for the compensation of distortions resulting from the effect of a laser-based treatment and a deep rolling, was calibrated by experimental data. The numerical design of these strategies provided a model for predicting compensation parameters to minimize profile grinding distortions.

**Keywords:** profile grinding; laser treatment; deep rolling; distortion

## 1. Introduction

Distortions occur frequently in manufacturing processes in industry. Thermally-induced distortions in particular are difficult to predict and control. During the processing of slim, long components, this has a serious impact on quality. Linear guides are an example of these.

A V-groove, which is usually embedded in such a component, increases the workpiece surface on one side, on which thermal edge zone influences arise during subsequent grinding processes to actually increase the workpiece quality. During machining, high loads, which lead to unwanted distortions in slim steel components, occur [1]. Apart from this, the production consists of a complex composition of sub-processes. Undesirable properties due to individual thermomechanical influences are the result. Post-process straightening processes are required due to the high demands with regard to the geometrical precision of a workpiece [2].

Deformations generated during grinding are mainly compensated iteratively and manually by procedures such as flame straightening. The methods from industry are increasingly being expanded to include automated processes. For this purpose, lasers provide a suitable heat source [3].

In the literature, however, the approach of local expansion by applying the rolling procedure is also increasingly proposed for residual stress and distortion reduction [4]. Locally induced stresses can thus counteract the effects caused by profile grinding. The challenge of a strong profile and dimensional fluctuations in the production of profile-grinded workpieces justify the development of a system for distortion compensation. Thus, this process potentially contributes to the economic efficiency of the entire process chain.

The aim of this research work was to predict grinding distortions of slim workpieces depending on the process parameters and to identify suitable thermal and mechanical straightening processes by laser treatment and deep rolling, which allow a specific compensation of the deformations.

The superposition of heat transfer and engineering mechanics problems and their effects on distortion could be analyzed by modeling the influences of the identified process parameters. The discussion of the effectiveness of scientifically investigated distortion strategies formed the basis for this. Finite element (FE) simulation methods were used to predict the boundary zone effects of the different processing steps on the surface and subsurface areas of the workpieces. By reproducing these effects to a sufficient extent and comparing them with experiments, a valid way of predicting the distortions that occur during the profile grinding of linear guide rails was developed [5]. With this method and the FE-based process know-how thus expanded, it was then possible to implement suitable measures for a thermal and a mechanical compensation method. The flexible applicability of a laser and a deep rolling tool in manufacturing process chains thus creates an opportunity for cost-effective distortion compensation.

In the following, the theoretical approach will be discussed first. This is divided into a description of thermally induced distortions of workpieces during grinding and laser machining and of mechanical influences on workpieces during deep rolling. For both problems, numerical approaches from the literature are also described. This is followed by the method section, which defines the approach and the parameters used in the three processes, consisting of grinding, laser machining and deep rolling. Section 4 shows the modeling strategy with the boundary conditions used for the finite element method. The results of the experimentally validated numerical simulation are used in Section 5 to predict grinding distortions and optimize compensation parameters. At the end, a short summary and a discussion on the added value of both compensation approaches follows.

## 2. Theoretical Approach

### 2.1. Thermally Caused Distortions

During the grinding of a steel surface with a V-groove, a thermal edge zone influence occurs [6]. This leads to an induction of tensile stresses near the surface, dominating on one side. In the case of slender workpieces, this results in significant workpiece distortions [7]. The thermally dominating process effects can be counteracted by reduction methods of an equivalent nature, produced on the opposite side of the V-groove.

One way of selectively inducing additional tensile residual stresses through a straightening process is based on global thermal expansion. Guan et al. first patented this invention, in which they used stationary heat sources and sinks [8]. This causes additional tensile residual stresses behind the heat source center. These stresses occurred after the workpiece cooled down with a resulting increase in the tensile zone.

According to Pilipenko, four parameters are decisive for the effectiveness of a thermal distortion reduction method as described by Guan et al. [8,9]. These are the maximum temperature in the contact zone, the width of the heated area, the heating time and the geometry of the workpiece with the positions of the original residual stresses to be counteracted. One of the main problems is that yield strength may be exceeded due to the induced compressive stresses, which could reverse the distortion effect. In addition to determining the optimal process parameters and maintaining the required temperature profile, the clamping system also influences the heat balance. Phase transformations in the material must also be taken into account [10].

Following Guan et al., Michaleris et al. described a further development of thermal distortion control by transient heat sources [8,11]. In addition to reducing the complexity of the process, it also increased the energy efficiency. The number of possible configurations for the parameter settings was increased. Residual stresses for distortion reduction could be introduced in a more targeted manner. Due to the principle of equilibrium, the generation of local compressive residual stresses also causes tensile residual stresses at other locations. The biggest disadvantage of the second type is embrittlement on the machined surface.

In recent publications, mainly WAAM-manufactured components and their plastic deformations have been analyzed [12,13]. The regular thermal cycles during coating cause process-related plastic strains. After unclamping the workpiece, intolerable deformations

are released. Fatigue and brittle fracture are the consequences of these thermally induced tensile residual stresses. With reference to the geometric precision, they are also caused by extensive deformation [14–17].

Numerical tools such as the finite element method (FEM) are suitable for determining the temperature, residual stresses and distortion [18–20]. The complexity from coupled mechanical and thermal submodels requires fine meshes and small time steps due to the strong nonlinearity. An efficient modeling approach with a low processing time is therefore only achievable for simple workpiece geometries [21].

For example, there are many such implementations for laser processing. The possibility of adaptive meshing for automated refinement in the machining area of the workpiece was helpful. Thus, the number of elements of the thermal–mechanical model could be reduced for a faster simulation [20,21]. Depending on the geometry and process to be represented numerically, according to Camilleri et al., a transformation of 3D problems into 2D problems could also be beneficial [22]. However, the effects of heat flows in the longitudinal direction could not be taken into account.

As a boundary condition in the FE method, Zhang and Michaleris succeeded in replacing a conventional Lagrangian model with an Eulerian framework, thus simulating a stationary heat source [23].

Coupled with an inherent strain model, Michaleris et al. described an efficient mechanical analysis method for predicting deformations [24]. For their mathematical formulation, they used an elastic analysis, taking the sum of the non-elastic strains as the inherent strain.

*2.2. Deep Rolling*

In the same way as using targeted thermal machining to counteract the residual stresses caused by grinding, deep rolling has also shown its effect as compensation method. When conducted on a surface, local stretching occurs perpendicular to it. The subsequent plastic deformation and expansion transverse to the loading direction leads to an induction of compressive residual stresses [25–27].

The application of the resulting compressive load stresses on the surfaces rolled over next to the V-groove allows the ground workpieces to be bent in the direction opposite to the distortion in a controlled manner [7]. During machining with a deep rolling tool, specific parameters can be varied. These can be the ball diameter, the rolling force, the feed rate and the track position.

Deep rolling is usually integrated as a forming process and has a high productivity with short cycle times and a high cost efficiency. This manufacturing procedure is often used to achieve a higher mechanical load capacity. The effects on turbine blades have been discussed by Klocke et al. [4]. A resulting increase in the strength of these machined components with a simultaneous increase in durability reduces the constant risk of a mechanical deformation during aerospace operations. This increase in hardness was caused by the application of compressive load stresses [4].

Rolling also offers a number of other advantages in the processing of steel workpieces, such as surface hardening through a reduction in roughness [28]. These advantages are based on complex fundamental mechanisms, which are difficult to understand purely by experiments. For this reason, a numerical analysis is also suitable for this process, especially with regard to combinatorial implementation as a compensation strategy.

In the context of the present work, deep rolling in the form of flat longitudinal rolling was considered as a possible distortion compensation method after profile grinding. Shape deviations can result from side-induced compressive load stresses due to the local stretching in the contact zone of the tool [4]. Focused induction can thus be used to counteract the deformations caused by grinding.

A typical quenched and tempered steel such as AISI 4140 behaves elastically up to the yield point. When the yield point is exceeded, elastoplastic effects occur. The workpiece begins to deform plastically. In order to be able to machine a larger surface area, the number of rolling passes can be increased with a corresponding lateral offset.

Studies according to the literature on multi-stage rolling processes have shown that the resulting residual stresses can be analyzed over time. This is possible through a model-based subdivision of the material into several volume elements. Thus, it allowed analyses of not only the local load of the tool on the workpiece, but also the load states at the next-neighbor elements affected by residual stress states during different process time steps [29].

The evaluation of a suitable parameter range for controlled stress induction in grinded, V-groove-endowed workpieces as a compensation option for the grinding distortion that occurs has already been described in an earlier publication [7]. With regard to this application, the FE-based method was also suitable for modeling effects appearing in the workpiece during and after the processes have been carried out.

In the literature, there are also some approaches in this respect. For example, Gornyakov et al. investigated efficient modeling algorithms for rolling [30]. They stated that in terms of efficiency and accuracy, models with implicit solution algorithms are the most suitable ones. They also considered rolling for clamped workpieces [31]. For their modeling, they compared short implicit and explicit 3D transient models, as well as the 3D Eulerian steady-state model and the short implicit 2D transient model for simulating the multilayer rolling of WAAM components. Depending on the solution accuracy, the number of elements (or degrees of freedom) as well as the choice between the more accurate implicit analysis method and the more inefficient explicit analysis method for the rolling models was relevant for the FEM analysis. They also found that using an elastically deformable roller instead of an analytically rigid roller had little effect on the solution quality. However, the simulation time was reduced due to this model change [30]. At the same time, the 2D models were unable to capture the 3D deformation mechanism during rolling. Likewise, they could predict the distributions of longitudinal residual stresses and plastic strains most imprecisely [30].

## 3. Materials and Methods

In the following, a description of the material used will be provided and the three machining processes will be discussed. These consist of the profile-grinding process on the one hand and the two straightening processes (laser machining and deep rolling) on the other. A methodological approach, consisting of the validation of the FE simulation with the experiments as well as the Gaussian regression process, follows in Section 3.3.

### 3.1. Workpiece

Profiles similar to linear guide rails were used in this research work. These workpieces, made of AISI 4140 with dimensions of 23 mm × 38 mm × 250 mm (h × w × l), were milled. On one side, centered over the complete length of the workpiece, there was an embedded V-groove with the dimensions of 8.67 mm × 19 mm (h × b) and a radius of 2 mm in the base. The dimensions in a picture can be found later on in Section 4.3.1. To ensure a better comparison of all machined steel profiles, they were stress-relief annealed (quenching temperature of 200 °C, 55 HRC) before the experiments were conducted. Table 1 shows the proportions of the alloying elements.

**Table 1.** The chemical composition of the workpiece material (wt%).

|  | C | Cr | Mn | P | S | Si | Mo | Ni | Al | Cu | Sn | Ti | V | Nb | Fe |
|---|---|---|---|---|---|---|---|---|---|---|---|---|---|---|---|
| AISI 4140 | 0.40 | 1.03 | 0.82 | 0.01 | 0.02 | 0.24 | 0.18 | 0.13 | 0.01 | 0.18 | 0.01 | 0.01 | <0.01 | <0.01 | balanced |

### 3.2. Experiments

Experimental approaches from previous publications were picked to set up a fully comprehensive prediction model. Numerical models already existed for each of the three processes mentioned here: profile grinding, laser treatment and deep rolling [3,7,32]. The thermal and mechanical loads caused by the tools as well as the boundary conditions on the

workpiece are described in Section 4. These constraints were calibrated for the parameter sets described below by means of an experimental adjustment and can thus be coupled with each other as boundary conditions within one FE model. A subsequent regression model used a neural network to extend the distortion prediction for parameter ranges that were not investigated experimentally. In addition, a simulation of the process chain consisting of grinding, lasering and deep rolling was performed for specific values. The distortion and compensation prediction of this range of values was again extended by the regressive approach. In the following, the experimental procedures of the grinding as well as the compensation processes are listed.

### 3.2.1. Grinding

The pre-milled and stress-relieved steel workpiece was grinded in the V-groove using a Baystate 9A60H16VCF2 profile grinding wheel with a grit size of 60 lines per inch. Before stress-relief annealing, the V-groove was pre-grinded with a corresponding profiled grinding wheel to ensure a continuous contact zone between the workpiece and the tool during the distortion experiments. Dressing was performed before each experiment using a diamond dressing roll (dressing parameters: radial dressing infeed $a_{ed}$ = 30 µm, speed ratio $q_d$ = 0.7). For cooling and lubrication, an oil (Houghton, Cut-Max 906-10) was supplied through a central nozzle. This fluid reached a flow rate of 50 L/min. For the definition of the boundary conditions in the process zone in the numerical model, the grinding forces in the tangential and normal directions, as well as the grinding power, were measured. The parameter ranges used can be found in Table 2. The grinding wheel feed $v_{ft}$ as well as the depth of cut $a_e$ (perpendicular to the base of the machining, e.g., the clamping plane) were varied. The cutting speed, $v_c$, was held at 35 m/s. The grinding happened in only one single pass.

**Table 2.** Grinding parameters.

| Parameters | Symbols | Values |
|---|---|---|
| Cutting speed | $v_c$ | 35 m/s |
| Feed rate | $v_{ft}$ | 1500–12,000 mm/min |
| Depth of cut | $a_e$ | 75–800 µm |
| Lubricant flow rate | $v_{fluid}$ | 50 L/min |

### 3.2.2. Laser Treatment

An 8 kW fiber laser from IPG Photonics (ytterbium fiber laser) was used for the controlled induction of tensile residual stresses in the workpiece on the opposite side from the V-groove. A circular heat source was moved centrally along the entire length of the bottom. It had a diameter $d_l$ of 15 mm. The value ranges of the two varied parameters, the laser feed rate $v_l$ and the laser power $P_l$, are described in Table 3. This allowed flexible control of the heat flow into the workpiece.

**Table 3.** Laser parameters.

| Parameters | Symbols | Values |
|---|---|---|
| Laser source diameter | $d_l$ | 15 mm |
| Laser powers | $P_l$ | 1.7–3.1 kW |
| Number of total passes | $N_l$ | 2 |
| Laser feed rates | $v_l$ | 0.5–1.0 m/min |

### 3.2.3. Deep Rolling

Table 4 lists the parameter sets for deep rolling, the second approach to compensate for the tensile residual stresses from grinding. This deformation control is based on the induction of residual compressive stresses on the workpiece shoulders next to the V-groove, which counteracts the caused distortion. An Ecoroll HG 13 with a tool diameter $d_b$ of 13 mm

was used. The lateral feed $f_{dr}$ varied in four stages and the number of total tracks on the surface changed at the same time. A hydraulically generated pressure of 475 bar on the deep rolling tool produced a normal force $F_{dr}$ of about 4.2 kN. Both the tool diameter and the rolling velocity $v_{dr}$ of 1000 mm/min were also left constant to ensure better comparability.

**Table 4.** Deep rolling parameters.

| Parameters | Symbols | Values |
|---|---|---|
| Tool diameter | $d_b$ | 13 mm |
| Deep rolling force | $F_{dr}$ | 4.2 kN |
| Number of total tracks | $N_{dr}$ | 1–50 |
| Deep rolling velocity | $v_{dr}$ | 1 m/min |
| Lateral feed | $f_{dr}$ | 0–1 mm |
| Lubricant | – | 5% emulsion |

### 3.3. Methods

In contrast with the previous interactive workpiece distortion reductive methods, where the compensation is exclusively experience based, the design presented in the next section is intended to simplify the process control for the user. For this purpose, the determination of the optimal compensation parameters was automated by means of a computer-based procedure.

First, an FE simulation model of the profile grinding, which had been validated by experiments, was used as a reference. This model was then able to predict the workpiece distortion for any parameter combination of feed rate and depth of cut. At the same time, missing distortions were determined by a neural network with a regression algorithm. The presented approach considered quality only in terms of the final shape. For the measurement, 13 measuring points were identified longitudinally on defined lines on the workpiece surface on both sides next to the V-groove, which were monitored with a coordinate measuring machine before and after each process step. The workpiece distortion was caused by locally induced residual stresses in the V-groove. However, this distortion had global effects on the shape after the workpiece was unclamped. Therefore, it was an important task to reduce distortions by adapted processes consisting of laser machining and deep rolling.

A validated simulation model was also created for the two straightening processes, which provided data as input for the regression model (Matlab, Regression Learner Toolbox). All three validated numerical process models were subsequently converted to an overall model. Simulated processes connected in series provided distortion data for the final compensation. In order to reduce the workpiece deformation, it was necessary to simulate the machining-induced effects from all three machining processes and validate the model. The grinding process was conducted first, followed by a combination of both straightening processes to minimize the distortion. The simulated deformations served as input for the neural network (artificial intelligence (AI)), whereupon a compensation algorithm determined the optimal straightening parameters. These values can be given as set points to the real system, which can use them to ensure the minimization of the workpiece distortion. A trained Gaussian process regression with the Matern 5/2 kernel and a constant basis function (isotropic kernel, automatic scale and signal standard deviation) was used. The input parameters were the depths of the cut as well as the feed rates, and the output parameters were the peak-to-valley distortions.

A central role of this algorithm is its use as a powerful tool that interprets the complex thermomechanical processes during profile grinding and compensation, and influences them in a targeted manner. For each workpiece distortion, it can identify the straightening parameters for laser machining and rolling for local compensation. With regard to the method, the first step was to evaluate the workpiece distortion based on the process parameters. The second step involved deriving a suitable combination of both straightening processes and their parameters that minimize process-induced distortions.

## 4. Model Development

In the following modeling sections, the elements of the concept for the computer-aided control of grinding distortions discussed here will be described first. This chapter establishes the connection between the individual methods and shows their transfer to the overall model.

### 4.1. Modelling Strategy

The numerical model for the distortion prediction of the respective process steps contained a large number of thermomechanical effects, which were given in the form of boundary conditions. The 3D FEM model consisted of two parts. At first, a transient heat transfer analysis was performed. This analysis yielded the temperature field in the workpiece for defined heat fluxes at the surface into the material (Figure 1). The second part dealt with the mechanical component. The temperature field from the first step served as the input. From this, thermal expansions, residual stresses and distortions were determined. In addition to the tools used, the clamping of the workpiece and the cooling lubricant during grinding defined further boundary conditions.

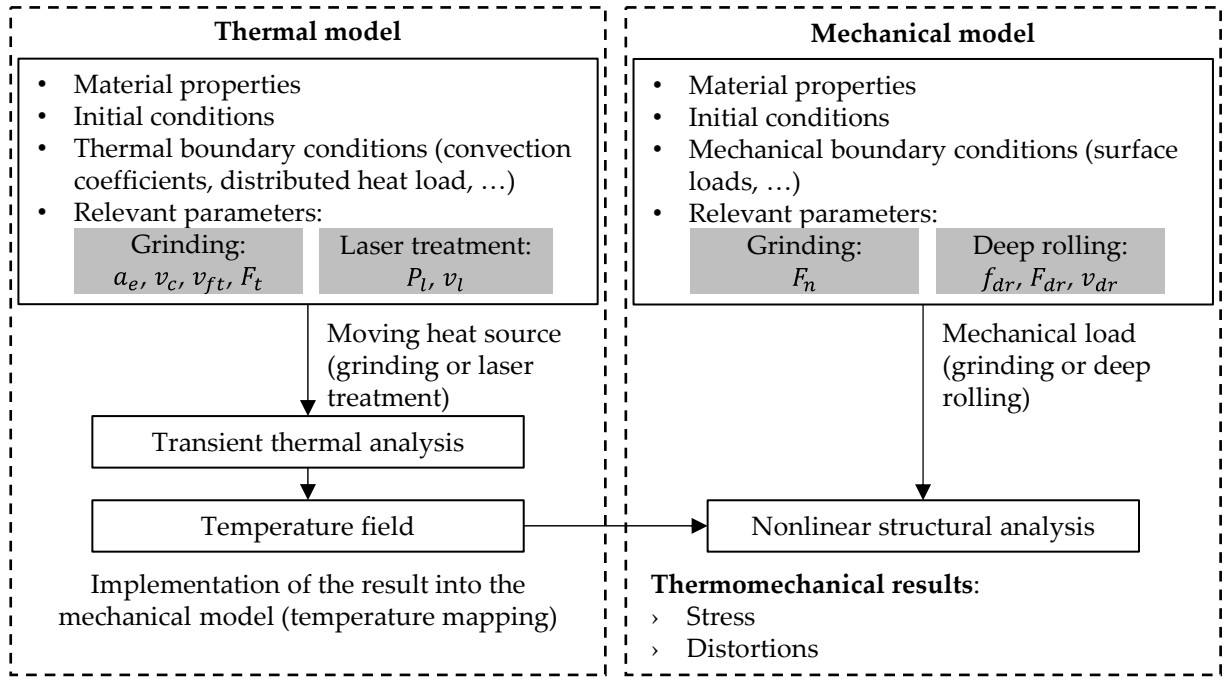

**Figure 1.** Thermomechanical finite element modelling strategy.

### 4.2. Material Properties

The material used was hardened, stress-relieved AISI 4140 steel. Table 5 lists the material properties at room temperature. These properties were used to solve the thermal and mechanical models.

**Table 5.** Thermal and mechanical properties of AISI 4140.

| Properties | Symbols | Values |
|---|---|---|
| Elastic modulus | $E$ | 210 GPa |
| Poisson's ratio | $\nu$ | 0.3 |
| Density | $\rho$ | 7830 kg/m$^3$ |
| Expansion coefficient | $\alpha$ | $1.15 \times 10^{-5}$ K$^{-1}$ |
| Specific heat capacity | $c_p$ | 450 W/(kgK) |
| Thermal conductivity | $k$ | 43 W/(mK) |

### 4.3. Thermal Model

#### 4.3.1. Heat Generation

The FE simulation of the temperature field was based on the FE solution of the heat equation:

$$\rho c_p \frac{\partial T}{\partial t} = \nabla \cdot (k \nabla T) + q \tag{1}$$

where $\rho$ is the density, $c_p$ is the specific heat capacity and $k$ is the thermal conductivity. The additional summand $q$ describes the used heat sources as the heat generation rate depending on the simulated process of grinding or laser machining.

For grinding, a triangular heat source for the simulation of the heat flux was found in the literature to be suitable for this purpose [5]. Its geometrical parameters depend on the locally variable contact length $l_g$ in the V-groove [7]. The heat flux itself depends on the tangential process forces $F_t$, which were measured via force platforms, and the energy partitioning factor $\varepsilon$ [33]. As a time-dependent boundary condition, this three-dimensional heat source with a triangular cross-section (see Figure 2) was moved along the V-groove during the simulation depending on the process parameters to be simulated.

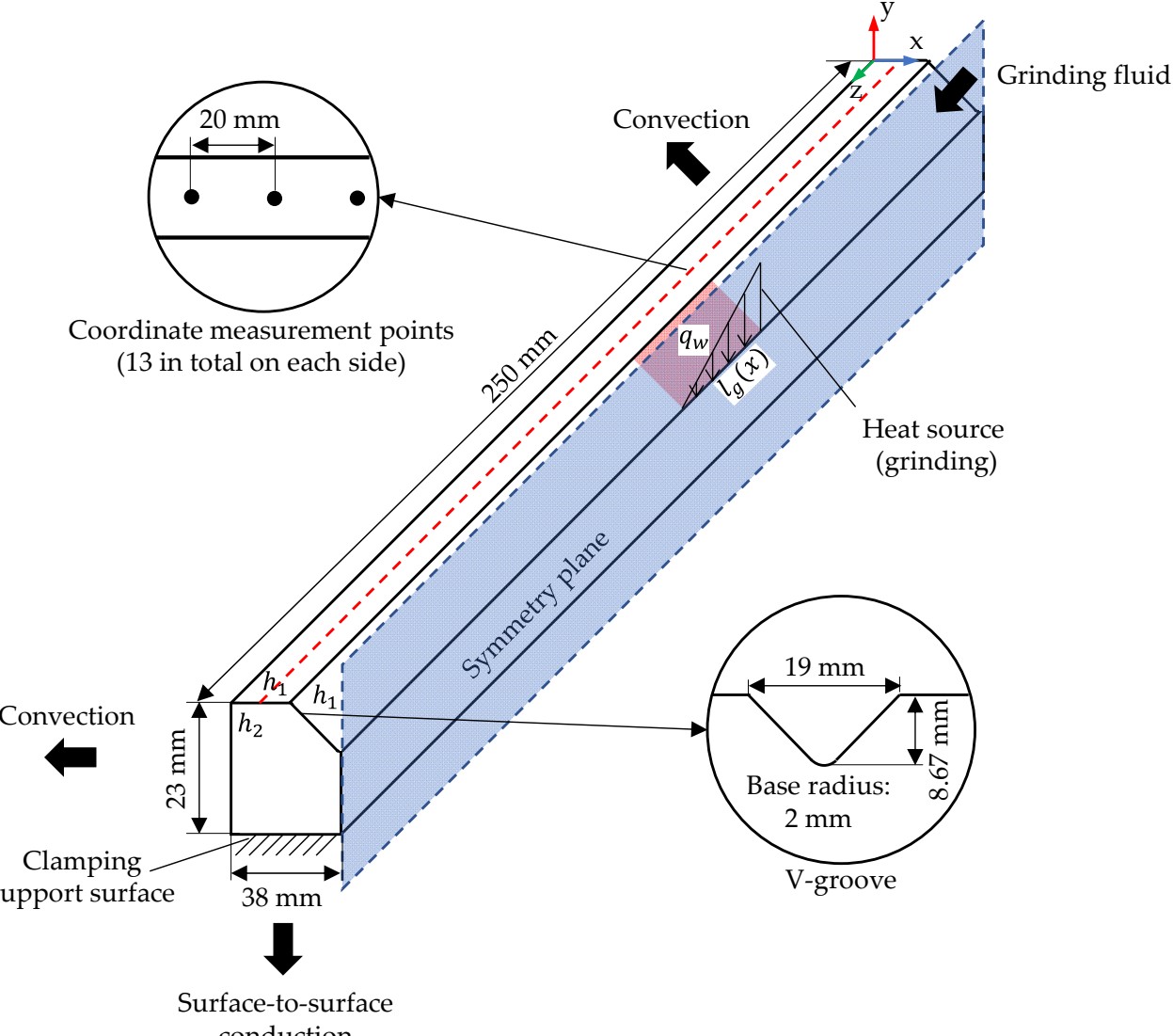

**Figure 2.** Workpiece set-up for the grinding process and the applied boundary conditions, with heat transfer coefficients and grinding contact zone for the simulation of the symmetric workpiece model.

For the thermal distortion compensation process, a further assumption was made in the mathematical model on the side opposite the V-groove. The representation of an annular laser source as a volumetric heat source was simulated as follows:

$$q = \frac{a \cdot P_l}{2\pi r_0^2 \left(e^{-(R/(2r_0))^2} + \frac{R}{r_0} \cdot \sqrt{\frac{\pi}{2}} \cdot \left(1 + erf\left(\frac{R}{\sqrt{2}\,r_0}\right)\right)\right)} \cdot e^{(x^2+y^2+R^2-2\cdot R\cdot\sqrt{x^2+y^2})/(2\cdot r_0^2)} \tag{2}$$

First, the absorption coefficient $a$ was set to 1 and then optimized correspondingly to the workpiece distortion. $P_l$ is the power of the laser. During the experiments and thus also during the simulation, the radius $R$ was defined as $d_l/2 = 7.5$ mm and the thickness of the ring $r_0$ as 2 mm. The global coordinates $x$ and $y$ were used with a time-dependent displacement of the heat source in the z-direction. The movement of the heat source for different sets of parameters was defined by setting a trajectory with the start point at one end of the bottom of the workpiece and with the end point at the other. For this line, a time range was given from the specified laser feed rate $v_l$. Double laser processing was conducted on this trajectory with an intermediate cooling interval of 170 s.

The thermophysical properties of the AISI 4140 steel were considered to be temperature-dependent. These properties were calculated numerically by taking into account the chemical composition. Thus, by curve-fitting at each time step and for each node, a function of the corresponding thermal conductivity ($k$), density ($\rho$) and specific heat capacity ($c_p$) was obtained. For the whole FE simulation, the workpiece model was simplified using the symmetry plane along the yz-plane to reduce the simulation time.

### 4.3.2. Thermal Diffusion

In addition to the process-related boundary conditions, heat losses to the environment due to convection and radiation were considered during the simulation. Assumptions for the heat transfer coefficients during the grinding process of $h_1 = 24 \times 10^3$ W/(m$^2$K) at the side surfaces and $h_2 = 6 \times 10^3$ W/(m$^2$K) in the V-groove were made based on the cooling lubricant according to Hadad's formulas [34]. Via a nozzle centered above the V-groove in front of the grinding wheel, the lubricant was directed onto the contact zone at a high volumetric flow rate. This resulted in forced convection with strong turbulent flows within the V-groove. At the lateral workpiece surfaces, the flow rate was lower; thus, the reduced forced convection was set as a boundary condition there. Due to the clamping on a magnetic plate, a constant temperature of 20 °C was set in the simulation on the side opposite the V-groove (see Figure 3).

During the laser processing of the workpiece on the bottom side, the boundary conditions were changed. The clamping contact was located on the two shoulder surfaces next to the V-groove. On the remaining surfaces, convection occurred via ambient air due to the absence of a cooling lubricant. Newton's cooling law can be used to calculate the convection heat loss.

Heat conduction within the workpiece was taken into account for both processes via the defined, microstructure-dependent heat conduction coefficient for AISI 4140.

### 4.4. Mechanical Model

The thermal fields obtained in the numerical calculation were linked to the simulation of the mechanical fields, so that residual stresses and deformations could be estimated and analyzed. This was achieved by decoupling the thermomechanical problem. The nonlinear calculation of the residual stresses and deformations induced by the grinding and laser machining accounted for the incremental numerical solution of the relationship between the strain and the displacement.

In addition to the previously mentioned thermal boundary conditions, mechanical conditions were also defined for the simulations. These were necessary for the grinding process and the second distortion compensation by deep rolling.

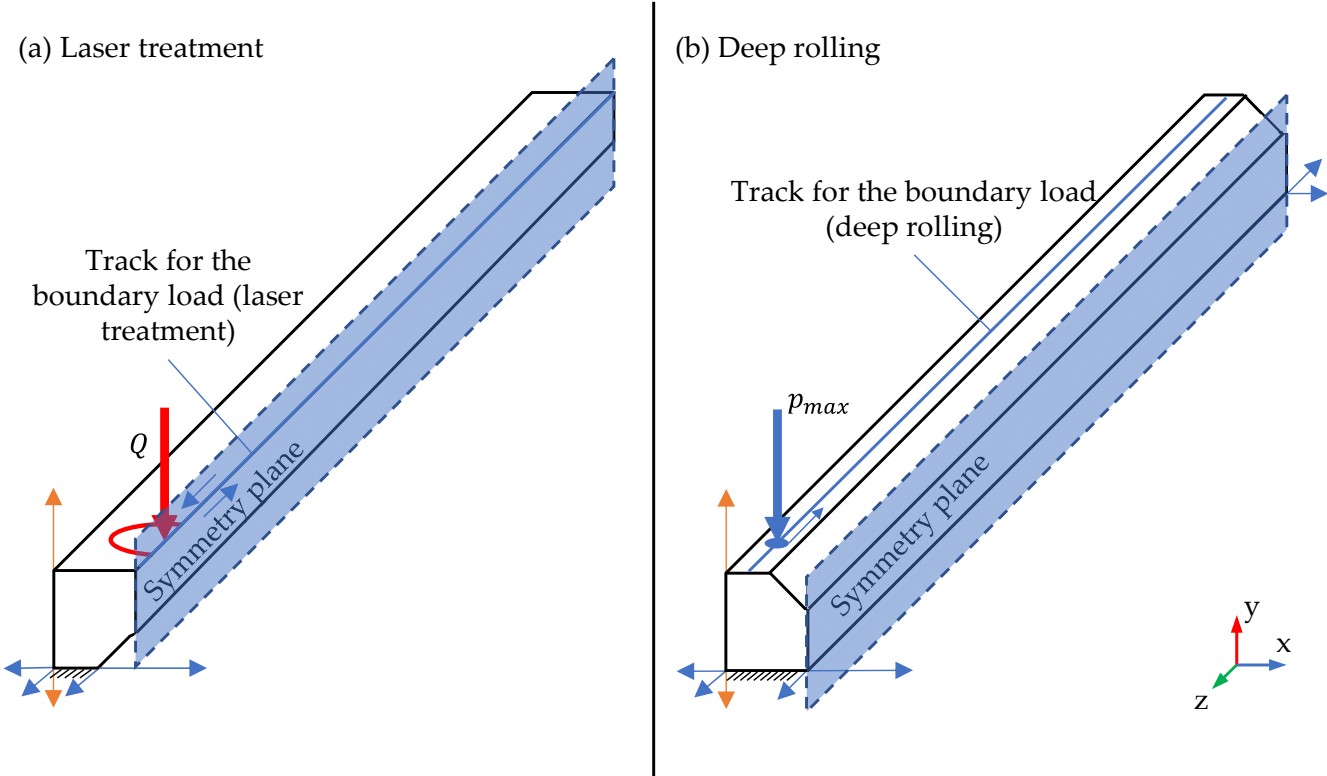

**Figure 3.** The applied boundary conditions for the two compensation processes: (**a**) laser treatment and (**b**) deep rolling for the simulation of the symmetric workpiece model.

In the contact zone between the grinding wheel and the workpiece, mechanical loads occurred in addition to the thermal loads. These loads were defined there as time-dependent surface loads for the individual parameter sets by the normal forces of the grinding wheel, which were measured during each experiment, as were the tangential forces.

The mechanical compensation process by means of deep rolling is similar. The interaction between a ball and a smooth surface in the case of rigid bodies corresponds theoretically to a point contact. In reality, however, there was a deformation of both components, which had to be considered in the simulation for the mechanical boundary condition. Due to the plasticity, the Hertzian model reached its limits and the calculation area in the contact zone in the plastic case changed to an approximately rectangular cuboid. This simplification was made as described below in the calibration of the workpiece model. The pressure area was defined according to Spinu et al. based on a defined exponential function [35]. Within the resulting contact area, a load of 85% was applied to approximate the compression area of the cuboid. The determination of the AISI 4140 material's parameters based on the manufacturing history of the workpiece allowed the plasticity to be modeled. Different loading levels, as described by Spinu et al. during a step-by-step rolling contact, were disregarded due to constant pressure and a uniform deep rolling velocity [35]. The feed rate per surface on the sides of the V-groove was approximated by a transient linear load activation. The possibility of track definition in the simulation from an interpolation of distinct start and end points could be exploited equivalently in grinding and laser machining.

The final readout of the resulting displacement from the thermomechanically coupled numerical simulation was compared with the experiments based on the peak-to-valley values. These values are defined as the height difference in the y-direction between a workpiece's center and end. They were used to calibrate the three submodels so that they could be coupled for the subsequent process chains of distortion compensation.

*4.5. Mesh*

The meshing of the workpiece in the simulation was conducted according to the sub-model and its boundary conditions. The thermal and mechanical loads made it necessary to optimize the results along the contact surface trajectory in order to more accurately calculate the numerical results close to the load zones. In a cross-sectional plane, a quad-mesh was defined with strong refinement towards the significant surfaces (the V-groove, the top and the bottom) and coarser meshing towards the interior of the workpiece, as well as towards the lateral surfaces. An extrusion of this two-dimensional model along the z-axis enabled a three-dimensional evaluation.

## 5. Results and Discussion

This chapter now presents the results of the FE model validation with the previously described experiments. The approach with AI allowed the extended prediction of the numerically and experimentally determined workpiece distortions after profile grinding. After that and after enabling the distortion prediction, the extended consideration by the two compensation processes follows. In Section 5.2, two compensation process chains as well as the optimized compensation parameters are presented, which were suitable for minimizing the distortion of the workpiece.

*5.1. Model Validation and Distortion Prediction*

Figure 4 shows the distortion values that resulted from the initial profile-grinding process. The experimental $\Delta PV$ values have been plotted on the left side of the figure for three selected feed rates from 1500 to 6000 mm/min with a variation in the depth of the cut from 75 to 800 μm. An increase in distortion was visible at a higher $v_{ft}$ with a simultaneously lower $a_e$. The curves were limited by an initial occurrence of grinding fires during the tests. While at a feed rate of 6000 mm/min, a maximum distortion of 0.04 mm occurred with a depth of cut of 200 μm, and at a $v_{ft}$ of 1500 mm/min, a value of 0.23 mm was reached with an $a_e$ of 800 μm. This maximum value provided the highest distortion for the subsequent straightening processes, which had to be compensated. The experimentally determined distortions were extended by the regression model, which acted as a simplified prediction model (AI) for the non-measured values. For the extended parameter range with feed rates up to 12,000 mm/min and respective depths of cut from 25 to 800 μm, the extended experimental and regression $\Delta PV$ values can be seen on the right side of the figure. Here, too, the early attainment of a maximum distortion for the respective feed rate at continuously lower depths of cut was visible. The experiments served as a data basis for the validation of the previously presented grinding simulation model. The numerically calculated distortions based on the simulated tensile residual stresses in the V-groove could also be seen for selected feed rates and depths of cut in Figure 4 (FEM). The minimum and maximum differences were 1% for the 700 μm depth of cut and 8% for 600 μm.

The experiments and the calculation approaches considered both the unprocessed workpiece without distortions and the ground workpiece to validate the models. Before and after each process step, the distortions were measured and compared with the simulation. For each of the three processes, consisting of grinding, laser machining and deep rolling, undeformed workpieces were used to validate the partial models. For the subsequent straightening processes, only corresponding workpieces that exhibited positive peak-to-valley distortion were considered. Using a selection of exemplary grinding parameter sets, the distortions were analyzed, the Gaussian regression model was trained and the numerical model was refined. On the numerical side, the process parameters of the compensation steps for each set of grinding parameters were simultaneously optimized until the positive distortions were within the error tolerance limit of 0.002 mm. Thus, negative deformations were only achieved for the deep rolling and laser machining processes, and were considered separately in the validation. In the case of the straightening processes, only deformed workpieces were assumed, for which their distortions were iteratively minimized towards zero subsequently.

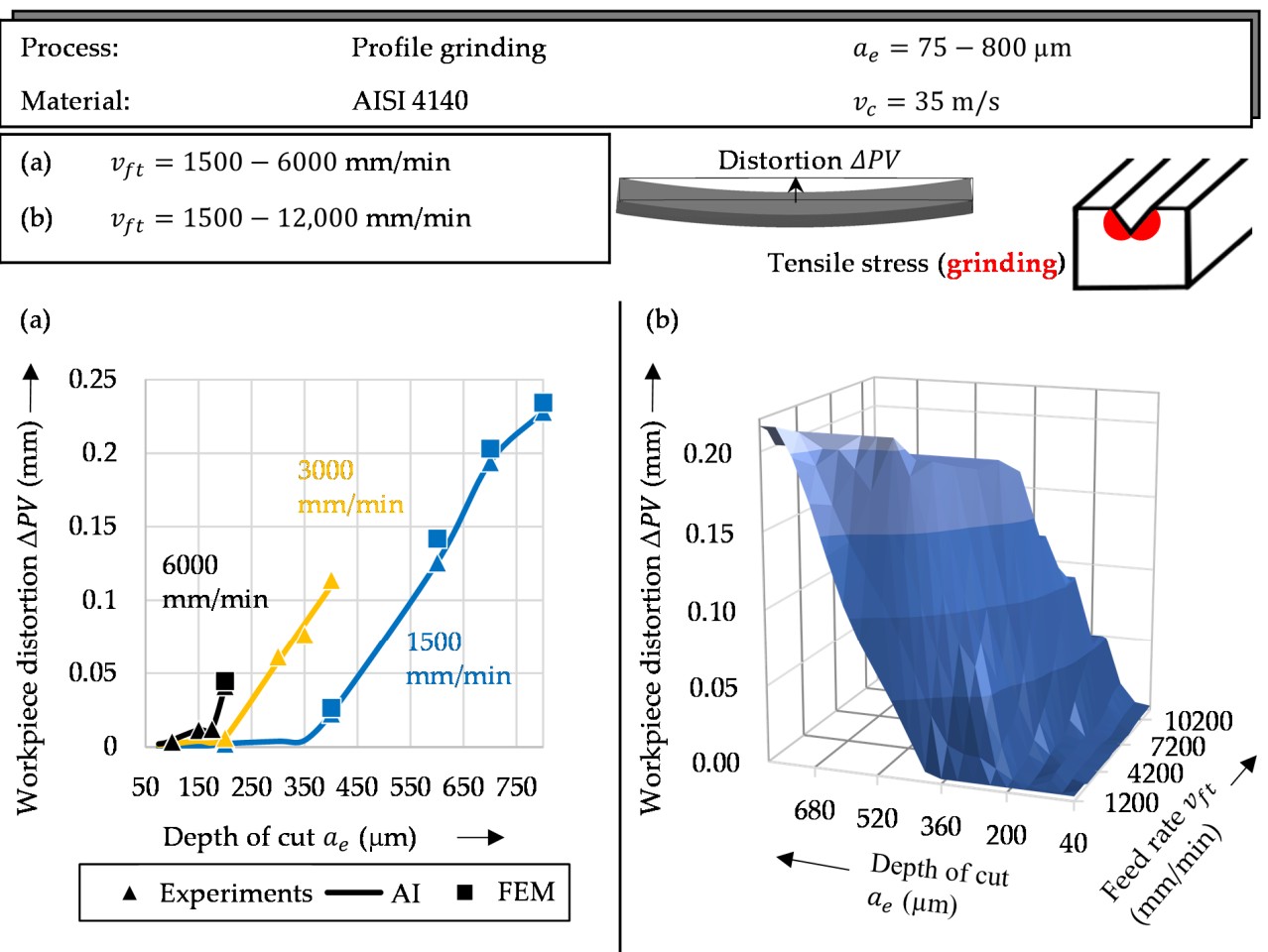

**Figure 4.** The workpiece distortion $\Delta PV$ (peak-to-valley) values after the profile-grinding process for different depths of cut and tangential feed rates: (**a**) examples for experimental and numerical results (FEM) and (**b**) distortions predicted by the AI regression model.

The experiments involving the laser-based process for straightening subsequent to grinding are shown in Figure 5. The left side shows the $\Delta PV$ values for the thermal processing of the side opposite the V-groove for four different laser feed rates from 0.6 to 0.9 m/min and a variation in the laser power from 1.7 to 3.1 kW. Four experiments were performed for each of the rates shown here, which in turn were interpolated by the Gaussian regression model. An increase in distortion could be seen with a reduction in the laser feed rate and with an increase in the laser power. At higher power values, however, a slower development of the $\Delta PV$ values could be seen. This result is also shown in the right-hand diagram for a larger parameter range. Only when a certain parameter combination was exceeded, the first distortions into the negative $\Delta PV$ range could be observed. A flexible continuous adjustment of the laser feed rate and power with maximum form deviations of $-0.27$ mm at 0.6 m/min and 3.1 kW makes this process suitable for straightening the grinding distortions. The combination of the profile-grinding process and the following laser processing has already been explained in sufficient detail in previous research work [3]. For this process, too, it was possible to determine $\Delta PV$ values with a maximum error of 7% by means of the FE simulation and the heat source mapped therein for the double pass ($N_l$) on the underside of the workpiece.

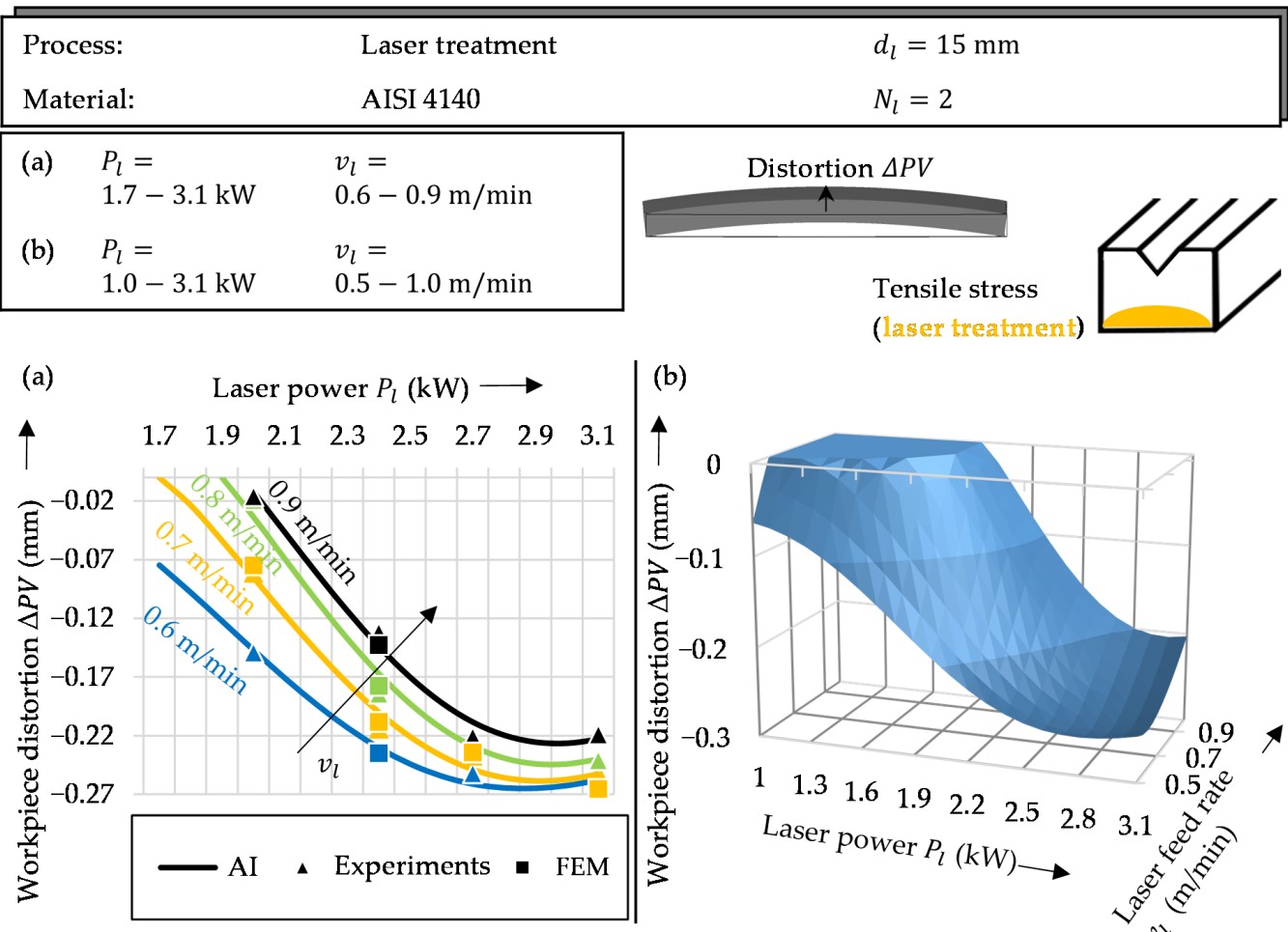

**Figure 5.** The workpiece distortion $\Delta PV$ (peak-to-valley) values after multi-pass laser treatment for different laser powers $P_l$ and laser feed rates $v_l$: (**a**) examples for experimental and numerical results (FEM) and (**b**) distortions predicted by the AI regression model.

As a second process, deep rolling was selected for the targeted introduction of compressive stresses on the surfaces next to the V-groove. In the tests required for this, only the number of tracks per side was varied with a corresponding offset of 0 to 1 mm. This allowed a discrete increase in the distortion of the workpiece resulting from the residual stresses. Figure 6 shows in the diagram on the left the experimentally determined distortions together with the results from the calibrated simulation model. While a $\Delta PV$ value of $-0.023$ mm could be determined with only one rolling pass per V-groove side, this value was already $-0.115$ mm with ten passes. The distortion not shown in the diagram, but measured in an experiment for 50 tracks, was $-0.123$ mm. Due to the small deviation from the $\Delta PV$ value for $N_{dr}$ equal to 10 with a simultaneously high time requirement, this value was used exclusively for calibrating the simulation model ($\Delta PV = -0.120$ mm for $N_{dr} = 50$), but was not taken into account in the further course of the work. Again, the regression model that was used provided reliable values for the calculation of the distortions for the other track numbers, with a small deviation from the experiments and the simulation.

*5.2. Prediction of Grinding and Compensation Distortion*

Based on the three simulated machining processes, consisting of profile grinding, laser machining and deep rolling, two process chains for distortion control were identified. The optimization was implemented by means of parameter adjustment. In the course of dynamic compensation, it was iteratively checked whether the target value (peak-to-

valley value) lay within the tolerance specifications of the experiments after the distortion simulation of the workpiece structure. If it was outside, the process was further optimized by changing the control parameters.

| Process: | Deep rolling | $d_b = 13$ mm | $F_{dr} = 4.2$ kN |
|---|---|---|---|
| Material: | AISI 4140 | $N_{dr} = 1 - 10$ | $v_{dr} = 1000$ mm/min |
| | | $f_{dr} = 0 - 1.0$ mm | |

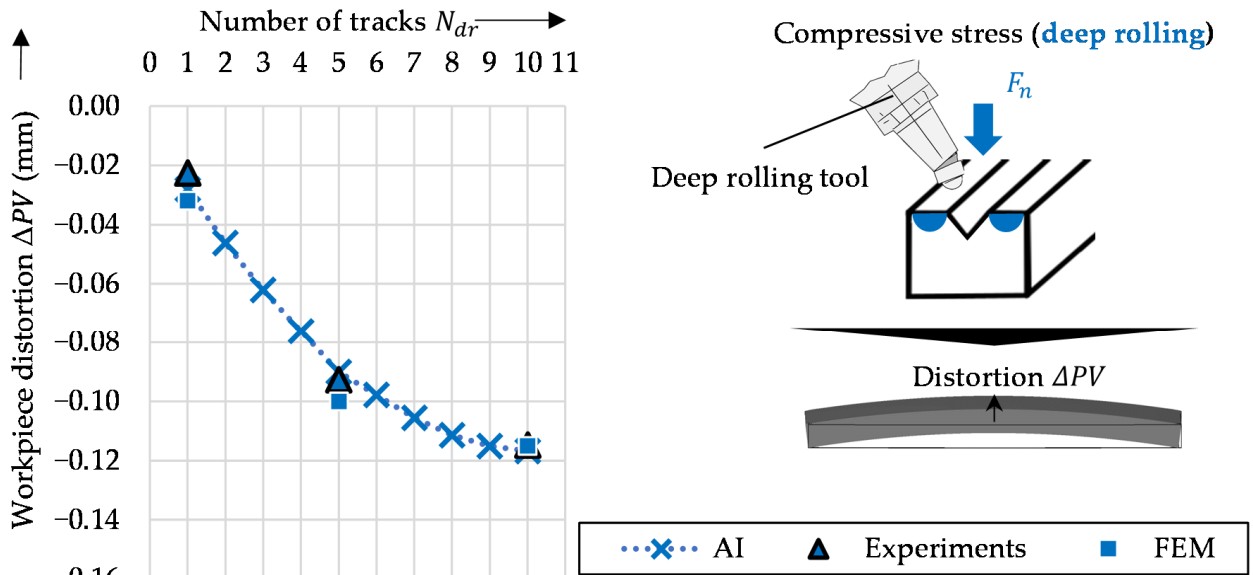

**Figure 6.** The workpiece distortion $\Delta PV$ (peak-to-valley) values after multi-pass deep rolling for different numbers of tracks $N_{dr}$: examples for experimental and numerical results (FEM) as well as distortions predicted by the AI regression model.

These two process chains were based on the possibility of the continuous selection of the two laser parameters (laser power $P_l$ and laser feed rate $v_l$) on the one hand, and discrete steps in the deep rolling tracks on the other. The distortion compensation by the first process chain is shown in Figure 7. For example, a profile-grinding process with a constant feed rate $v_{ft}$ of 1500 mm/min and a varying depth of cut $a_e$ between 150 and 800 μm was considered. This range showed the widest bandwidth of distortions in the experiments. At a depth of cut of 800 μm, a maximum peak-to-valley value $\Delta PV$ of 0.228 mm was observed. By means of the subsequent laser machining and the tensile residual stresses, thus introduced relative to the V-groove, it was possible to control the distortion. The model validated by the experiments offers different parameter combinations for the respective grinding distortions, consisting of the laser power and the laser feed rate.

As the experiments showed, deep rolling also enabled a distortion reduction. The limiting factor here was the discrete number of rolling tracks. These offered far less flexibility of application compared to the continuous adjustment capabilities of the laser. The material damage and severe microstructural transformations on the lasered surface could be reduced by reducing the laser power as well as increasing the laser feed rate. To make this possible, a subsequent machining combination of deep rolling and the following laser treatment was the logical approach.

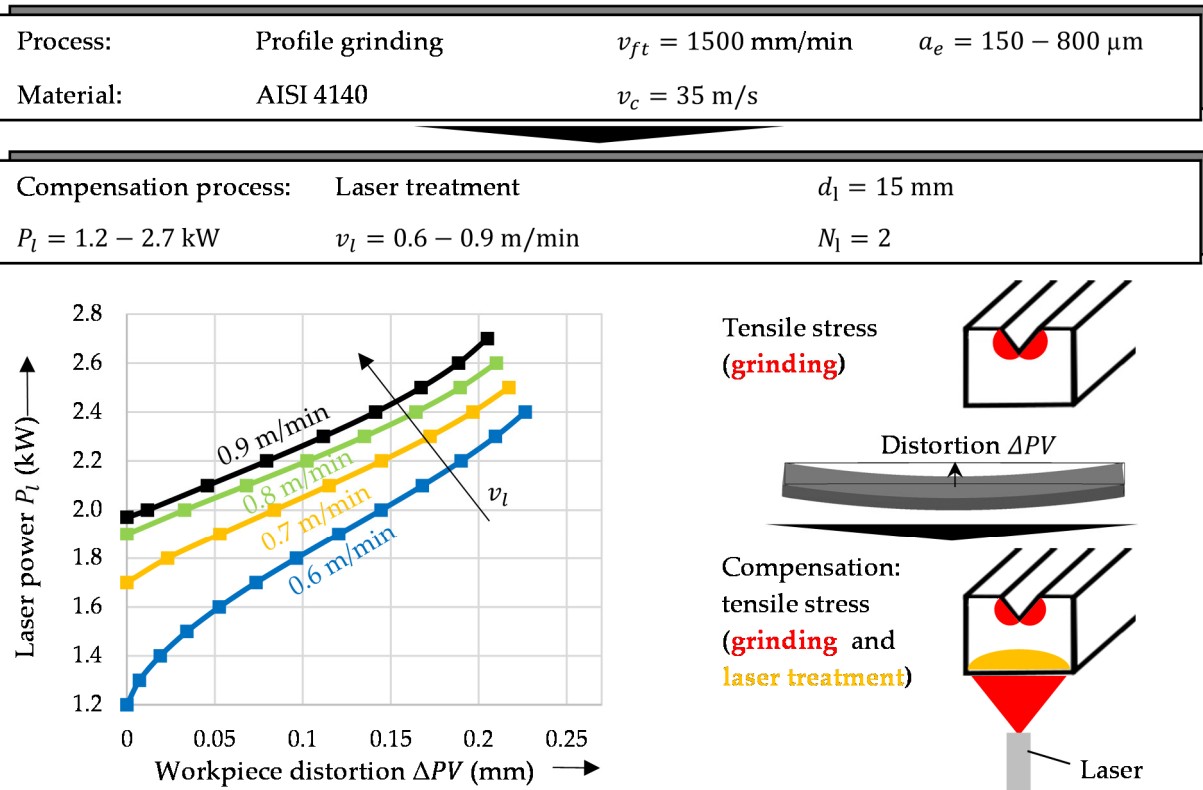

**Figure 7.** Distortion compensation strategy by multi-pass laser treatment with different laser powers $P_l$ and laser feed rates $v_l$ (optimal parameters) for workpieces ground with a constant feed rate of 1500 mm/min and depths of cut between 150 and 800 µm.

The final thermal compensation process had to be applied in each case due to the discrete parameter steps involved in the deep rolling process. Figure 8 shows the necessary parameters for distortion control for the exemplary grinding operations from the first process chain. On the left side, the reduction in distortions due to the discrete deep rolling tracks from one to ten is shown. For each particular peak-to-valley value, the diagram reflects the respective track number with the blue lines and the remaining distortions after the first compensation step with the yellow ones.

For example, if a workpiece distortion of 0.04 mm is obtained after grinding, this can be minimized with a maximum number of tracks $N_{dr}$ of one (blue line). The remaining residual distortion is shown by the yellow line in the left diagram. It amounts to 0.013 mm. This residual distortion can now be minimized in the second process step, laser machining, until it reaches the error tolerance of 0.002 mm. In the right diagram, Figure 8 shows the guide parameters for post-processing by laser. The necessary laser processing parameters for complete distortion minimization can be seen. The remaining distortion of 0.013 mm after deep rolling in the first process step mentioned in the example can thus be compensated for with a parameter combination of 1.29 kW and 0.6 m/min (blue line) or 1.72 kW and 0.7 m/min (yellow line). Depending on the selection of the feed rate, associated laser powers corresponding to the residual distortion of the workpiece were identified. Here, laser treatment was the final step in the process chain. It can be seen that, at a higher feed rate, the amount of heat missing locally on the underside had to be compensated for by a higher laser power in order to achieve the same compensatory effect. With a longer processing time at lower powers, the workpiece heated up into deeper layers. At a higher $v_l$ and an increased $P_l$, the direct surface was significantly affected and microstructural changes occurred.

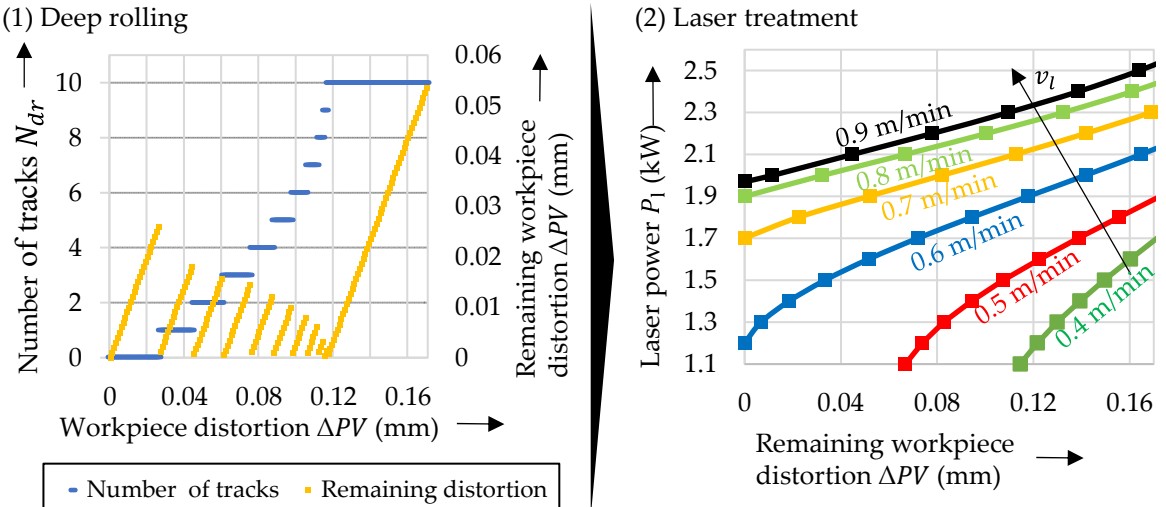

**Figure 8.** Distortion compensation strategy by multi-pass deep rolling (1) with a different number of tracks $N_{dr}$, leaving a distortion, and by a following laser treatment (2) for different laser powers $P_l$ and laser feed rates $v_l$ (optimal parameters) for workpieces ground at a constant feed rate of 1500 mm/min and depths of cut between 150 and 800 μm.

The residual compressive stresses led to a contraction of the bottom surface on the V-groove side. The tensile stresses of laser machining also deformed the workpiece in the same direction. The method used was based on the idea that the internal loads (residual stresses) that cause the deformation of the ground workpiece are balanced between its top and bottom surfaces by the subsequent processes so that the deformation is eliminated.

## 6. Conclusions

This paper presents a novel method for the compensation of workpiece distortions during profile grinding by means of laser machining and deep rolling. The method included first the numerical modeling of the three processes and then calibration by an experimental implementation. The 3D FEM model developed in this work is a method for the thermomechanical analysis of shape deviations when tensile residual stresses are induced by profile grinding and laser machining, and compressive ones are introduced by deep rolling. It is based on detailed definitions of the corresponding relevant surface loads and the boundary conditions, and predicts the distortions. With the consistency of an existent 3D FEM model and an AI model, sustained by experimental results, this paper provides important machining parameters for distortion compensation. The second part of the method is related to the identification of two process chains and an optimization for distortion reduction. The coupled valid submodels show that this method can be used to compensate for workpiece distortions. Currently, the method is based on numerical

prediction and optimization. Future work will consider the implementation of post-process distortion control for an automated industrial application to improve the accuracy of the method.

**Author Contributions:** Conceptualization, C.S.; methodology, C.S. and M.H.; software, C.S.; validation, C.S.; formal analysis, C.S.; investigation, C.S.; resources, C.S. and M.H.; data curation, C.S. and M.H.; writing—original draft preparation, C.S.; writing—review and editing, M.H., M.F.Z. and C.H.; visualization, C.S.; supervision, M.F.Z. and C.H.; project administration, C.S. and M.H.; funding acquisition, M.F.Z. and C.H. All authors have read and agreed to the published version of the manuscript.

**Funding:** This research was funded by the German Research Foundation (DFG), grant numbers HE 3276/9-1 and ZA 288/64-1, project number 402705371.

**Data Availability Statement:** The reference data presented in this study are available upon request from the corresponding author.

**Conflicts of Interest:** The authors declare no conflict of interest.

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
