# Peer review of "Combination of Thermal and Mechanical Strategies to Compensate for Distortion Effects during Profile Grinding"

_machines, doi:10.3390/machines10121240_

Round 1

Reviewer 1 Report

1. This paper presents a novel method for the compensation of workpiece distortions during profile grinding by means of laser machining and deep rolling. The method included first the numerical modeling of the three processes and then calibration by an experimental implementation. The 3D FEM model developed in this work is a method for the thermomechanical analysis of shape deviations when tensile residual stresses are induced by profile grinding and laser machining and compressive ones by deep rolling. As the experiments showed, deep rolling also enabled a distortion reduction. The limiting factor here was the discrete number of rolling tracks. These offered far less flexibility of application compared to the continuous adjustment capabilities of the laser. Overall, the paper structure is complete. The picture is clear and readable.

    2. It is recommended to add abstract content.

Author Response

Dear Reviewer,

Thank you very much for your comments. They were very helpful for the further development process of our publication. After a thorough review, we were able to integrate all of your technical comments with complementary research methods, definitions, and results. Through these additions, you have made an essential contribution to taking the paper to a new level of development. You not only made a fundamental contribution to our publication but your remarks and suggestions have also helped us to improve and expand our approach and expertise with regard to future research and journal articles.

We would like to ask you to review the following comments on the newly integrated research results and to reconsider your opinion.

Many thanks.

With kind regards

The Authors

Point 1: It is recommended to add abstract content.

Response 1: Thank you for the very good suggestion regarding the insertion of abstract content. I have implemented your suggestion and inserted them in appropriate places. The text is now more understandable for the reader and he can follow the contents better. Besides the extension of the introduction, there is now a short additional overview in each section of the text.

Reviewer 2 Report

Congratulations for this very valorous work!

Some minor explications can be added. All my remarks are in the attachment.

Author Response

Dear Reviewer,

Thank you very much for your comments. They were very helpful for the further development process of our publication. After a thorough review, we were able to integrate all of your technical comments with complementary research methods, definitions, and results. Through these additions, you have made an essential contribution to taking the paper to a new level of development. You not only made a fundamental contribution to our publication but your remarks and suggestions have also helped us to improve and expand our approach and expertise with regard to future research and journal articles.

We would like to ask you to review the following comments on the newly integrated research results and to reconsider your opinion.

Many thanks.

With kind regards

The Authors

Point 1: Chapter 3 describes the structure of the experiments. I suggest here to clarify the nature of the profile grinding operation. The rounded V profile is machined with a corresponding profiled grinding wheel, with a cutting contour extended on the whole profile, or it is machined side-by side, and finally the rounded bottom of the groove?

Response 1: Before stress-relief annealing the V-groove was pre-grinded with the same grinding wheel to ensure a continuous contact zone between workpiece and tool during the distortion experiments. Now, I mentioned that in the paper

Point 2: The table 2 and the text before indicates the parameters of the dressing of the grinding wheel. Till the goal is to obtain a reconditioned tool with the properties as before (excepting the outer diameter evidently), I consider that these parameters, namely the radial dressing infeed ad and the dressing speed ratio qd could be omitted from the table because they don’t influence the cutting parameters applied.

Response 2: You are absolutly right! They have no influence. I removed them from the table.

Point 3: As it results from the table and the description given before, the depth of cut ae [75, 800] µm is given perpendicular to the base of machining e.g. the clamping plane or normal to the profile?

Response 3: ae is perpendicular to the clamping plane. Now, I specified it in Chapter 3.2.1.

Point 4: The depth of cut given here equals the machining allowance, meaning that the grinding process results through one single pass, or there are multiple passes over the surface? If so, indicate please the number of passes. If not, I consider it should be mentioned that the grinding happens in only one single pass.

Response 4: It was always only one single pass. Now, I specified it in Chapter 3.2.1.

Point 5: Chapter 5 presents the results of the combined approach: experimental, FEM and AI. Here the deformations caused by the grinding process are positive, while those obtained by laser or deep rolling are negative. For me it seems logical that the final goal is to achieve the zero deformation value. The experiments and the computational approaches have considered the machined/grinded workpiece that is already deformed or a workpiece with V-groove without deformations? If this latter, how was that straightened? If not, it seems that positive deformations turn into negative deformations, and surely isn’t the goal. Here You should give some clarification.

Response 5: The experiments and the calculation approaches have considered both the unprocessed workpiece without distortions and the ground workpiece to validate the models. Before and after each process step, the distortions were measured and compared with the simulation. For each of the three processes, consisting of grinding, laser machining and deep rolling, undeformed workpieces were used to validate the partial models. For the subsequent straightening processes, only corresponding workpieces that exhibited positive peak-to-valley distortion were considered. Using a selection of exemplary grinding parameter sets, the distortions were analyzed, the Gaussian regression model was trained and the numerical model was refined. On the numerical side, the process parameters of the compensation steps for each set of grinding parameters were simulatively optimized until the positive distortions were within the error tolerance limit of 0.002 mm. Thus, negative deformations are only achieved for the deep rolling and laser machining processes considered separately in the validation. In the case of the straightening processes, only deformed workpieces were assumed for which their distortions were iteratively minimized towards zero subsequently. I clarified it now. Thank you for the remark.

Point 6: Lines 506-508: “On the left side, the reduction of distortions due to the discrete deep rolling tracks from 1 to 10 is shown. For each particular peak-to-valley value, the diagram reflects the respective track number with the blue lines and the remaining distortions after the first compensation step with the yellow ones”. This explanation is not clear. Please, give here an example, for making the graph understandable. There exist a blue horizontal line starting from zero to 0,03, and a correspondent yellow oblique line, being the diagonal of the domain [0, 0,03] x [0, 5]. How to interpret this, and the other similar pairs?

Response 6: I included now an example with a starting workpiece distortion of 0.04 mm after grinding and explained a step-by-step solution. The grinding distortion can be minimized with a maximum number of tracks N of 1 (blue line). The remaining residual distortion is shown by the yellow line in the left diagram. It amounts to 0.013 mm. This residual distortion can now be minimized in the second process step, laser machining, until it reaches the error tolerance of 0.002 mm. In the right diagram, Figure 8 shows the guide parameters for post-processing by laser. The necessary laser pro-cessing parameters for complete distortion minimization can be seen. The remaining distortion of 0.013 mm after deep rolling in the first process step mentioned in the mentioned example can thus be compensated with a parameter combination of 1.29 kW and 0.6 m/min (blue line) or 1.72 kW and 0.7 m/min (yellow line).

Point 7: Lines 509-510: “On the right side of the figure, the necessary laser processing parameters for complete distortion minimization can be seen” What is meaning ‘the complete distortion minimization’? What is the minimal distortion value that is invoked here? From any residual deformation shown in the abscissa, a vertical that intersect the traces, shows the necessary laser power value, that must be applied by the indicated feed rate, to achieve the minimal deformation ? Please give some additional explications, like I suggested before.

Response 7: I continued the example and provided a step-by-step solution mentioned in your Point 6 for the laser process. The minimal deformation (acceptable error of the compensation) is 0.002 mm. Thanks for pointing out that a detailed example makes things clearer for the reader.

Point 8: Lines 533-535: “The 3D FEM model developed in this work is a method for the thermomechanical analysis of shape deviations when tensile residual stresses are induced by profile grinding and laser machining and compressive ones by deep rolling.” Rather this very valorous paper shows also the consistency of an existent 3D FEM model and an AI model, sustained by experimental results, and provides very important machining parameters. You should emphasize that, because it is much more that a FEM model development.

Response 8: Good point! Now, I pointet that out in the conclusion. Thank you very much. It provides a better overall summary of our work.  

Point 9: The English is, in my opinion, good, but I have to specify I’m not a specialist in English. However, in some places the formulation can be simplified for a better understanding. I indicate in detail in the next section.

Response 9: Thank you very much for the detailed list of suggestions. They were very helpful and improved the language as well as the reading flow significantly. I included all of them!